

# Impact of foam rolling with and without vibration on muscle oxidative metabolism and microvascular reactivity

Haizhen Huang, Bin Leng and Chuan Zhang

Central China Normal University, Wuhan, Hubei, China

## ABSTRACT

**Background and Purpose**. There is a growing interest in use vibration foam rolling as a warm up and recovery tool. However, whether vibration foam rolling offers additional benefits to traditional foam rolling is unclear. The current study aims to compare the effects of acute foam rolling, with and without vibration, on skeletal muscle metabolism and microvascular reactivity.

**Methods**. Fifteen physically active young males were tested on two different days, with gastrocnemius muscle microvascular function assessed using near-infrared spectroscopy coupled with the post-occlusive reactive hyperemia technique, before and after foam rolling, performed with or without vibration. The slope of tissue saturation index (TSI) decrease during occlusion between 120 s to 150 s (TSI occlusion slope) was assessed for muscle metabolic rate. Three commonly used microvascular function indexes, including the first10s TSI slope after occlusion (TSI10), time for TSI to reach half of peak magnitude (TSI1/2), and TSI peak reactive hyperemia, were also assessed.

**Results**. None of the measured indexes showed significance for interaction or method (all $p > 0.05$). However, there was a main effect for time for TSI occlusion slope, TSI1/2, and TSI peak reactive hyperemia ($p = 0.005$, 0.034 and 0.046, respectively). No main effect for time for TSI10 was detected ($p = 0.963$).

**Conclusions**. The application of foam rolling can decrease muscle metabolism, and may improve some aspects of muscle microvascular function. However, vibration foam rolling does not seem to offer any additional benefits compared to traditional foam rolling alone.

## INTRODUCTION

Foam rolling is a popular self-myofascial release technique that has been increasingly adopted in exercise and clinical settings due to its simplicity and affordability. Foam rolling generally involves applying the user's own body weight on a foam roller, which puts pressure on specific muscles. While controversies do exist (*Healey et al., 2014*; *Jones et al., 2015*), there is evidence that the application of foam rolling is able to enhance muscle recovery, joint range of motion and overall performance (*Cheatham et al., 2015*; *Wiewelhove et al., 2019*). The mechanisms through which foam rolling exerts its effects have been extensively investigated. Some suggest that the pressure applied on the fascia may break the adhesions

Corresponding author
Chuan Zhang,
chuanzhang@ccnu.edu.cn

and knots of the fascia (*Wiewelhove et al., 2019*). Neurologically, it was demonstrated that foam rolling intervention may modulate pain perception (*Aboodarda, Spence & Button, 2015*; *Pearcey et al., 2015*), and reduce muscle tension from stimulating the Golgi tendon (*Bradbury-Squires et al., 2015*). Importantly, foam rolling may also induce changes in the circulatory systems (*Okamoto, Masuhara & Ikuta, 2014*). Studies have shown that foam rolling may accelerate the remove of lymphatic fluid and thereby reduce edema (*Shu et al., 2021*), and increase blood flow which helps with nutrients delivery and waste removal (*Hotfiel et al., 2017*).

Vibration is another commonly used intervention technique in sports and rehabilitation for muscle, which employs mechanisms similar to those of foam rolling. Both interventions are believed to exert its effects by acting on the proprioceptors (*Freiwald et al., 2016*; *Jordan et al., 2005*), and induce changes in peripheral circulation (*Robbins, Yoganathan & Goss-Sampson, 2014*; *Schroeder, Wilke & Hollander, 2021*; *Soares et al., 2020*). Due to the potential synergistic effect, there is a growing interest in combining the two together with the purpose of amplifying their benefits. Vibration foam rolling, where the foam roller vibrates during the application of tradition foam rolling, has therefore been studied extensively. Some suggested that vibration foam rolling, performed at both low and high frequencies, can improve knee flexion range of motion (ROM) up to 20 min following application (*Kasahara et al., 2022a*). One study has shown that the improvement in hip ROM by vibration foam rolling is significantly greater (*Reiner et al., 2021*). Another study demonstrated the pain pressure threshold increase and tissue hardness decrease lasted longer for the vibration condition compared to the traditional vibration condition (*Kasahara et al., 2022a*). However, there are also studies which showed no significant difference compared to traditional foam rolling (*de Benito et al. 2019*; *Romero-Moraleda et al., 2019*). Some researchers have also sought to investigate the combined effects of vibration foam rolling with other types of warm-ups, such as static stretching (*Chen et al., 2022*), indicating a potentially broad future application for this technique. Therefore, the precise physiological mechanisms and the comparative effectiveness of vibration *versus* non-vibration foam rolling requires further elucidation.

Recent studies have investigated the effects of foam rolling, performed on the upper extremity by another person, on muscle microvascular function and found acute improvements (*Soares et al., 2020*). On the other hand, compared to the upper body muscles, leg muscles are larger and predominantly involved in weight-bearing activities, which may require a more prolonged or intense stimulus to induce similar microvascular changes. Additionally, the leg have diminished vascular conductance (*Proctor, Le & Ridout, 2005*) and a higher proportion of slow-twitch muscle fibers (*Elder, Bradbury & Roberts, 1982*). However, whether similar effects can be observed at the lower extremities is currently unknown. Additionally, it is unclear whether vibration foam rolling may have a different influence on skeletal muscle microvascular reactivity compared to foam rolling alone. Given the increasing interest in using vibration foam rolling both for post-exercise recovery as well as a warm-up technique (*Li et al., 2023*; *Wang et al., 2022*), as well as the critical role of these physiological attributes during the process, it is therefore necessary to conduct comparative research to determine whether vibration foam rolling may provide further

health and performance related benefits compared to foam rolling alone, and to clarify the potential mechanisms.

In an effort to bridge this gap in the current knowledge regarding the physiological mechanisms underpinning the potential additive effects of vibration foam rolling compared to traditional foam rolling, this study is designed to evaluate whether foam rolling, performed with and without vibration, could induce beneficial oxidative metabolic and microvascular reactivity changes in the gastrocnemius in physically active young males. We are interested in these physiological aspects, as they are critical for optimizing muscle performance, enhancing recovery, and potentially improving overall exercise efficiency (*Barstow, 2019*; *Cavanaugh et al., 2017*; *Delp & O'Leary, 2004*; *Soares et al., 2020*). Findings from this study could be informative for the use of vibration foam rolling as a recovery as well as a warm-up tool, and contribute to a broader understanding on foam rolling's impact of muscle health and athletic performance.

## MATERIALS & METHODS

### Participants

Fifteen male participants who regularly engage in various physical activities, and had experience using foam rolling as part of the exercise routine were recruited for this study. This study recruited males only in order to avoid the potential influence of gender related difference on vascular function measures (*Green et al., 2016*; *Keller & Kennedy, 2021*). The inclusion criteria were (1) 18–30 years old, (2) generally healthy without any cardiovascular or musculoskeletal diseases, (3) regularly engaging in physical activity as least three times a week for 60 min per session, (4) BMI <30 kg/m$^2$. This study was approved by the Institutional Review Board at Central China Normal University (NO. CCNU-IRB-202306005a), and a written informed consent from all participants prior to any data collection.

### Study design

This is a single group, repeated measure design. The participant reported to the lab on two different days, separated by at least 48 h and a maximal of 7 days. On the first visit day, after taking height and weight measures, the participant lied on a yoga mat placed on the ground for at least 5 min. The skeletal muscle oxidative metabolism and microvascular function assessment were then carried out on the gastrocnemius muscle. Foam rolling intervention with or without vibration was carried out subsequently, and the assessments were repeated immediately following the intervention. The order for foam rolling (with or without vibration) was randomly assigned for each participant. The tests were repeated on the second visit, and the only difference was the foam rolling method (*i.e.,* change from with vibration to without vibration).

### Foam rolling

The foam rolling intervention was carried out using a cylindrical roller (Vyper 2.0; Hyperice, Irvine, CA, USA) that's capable of vibrating at three different intensities (48 Hz, 60 Hz and 72 Hz). For the vibration foam rolling intervention, the roller was turned on to

medium intensity (60 Hz). For the foam rolling intervention, the roller was turned off. The participant had a chance to familiarize with both types of intervention when first reported to the lab on the non-testing side, prior to the ultrasound assessment.

The foam rolling intervention was a two-minute protocol. Each roll starts from the popliteal fossa which gradually rolls down to just above the Achilles tendon, and then roll back. Each roll was completed in 4s. A metronome was used to help the participant control the rolling speed. During foam rolling, the participant was instructed to put his contralateral leg across the intervention leg, and place the body weight on the rolling side to apply the pressure on the roller. The pressure was controlled using a visual analog scale (VAS), which is a scale ranging from 0-10 with higher numbers indicating worse perceived pain. The VAS was controlled at 6–7, similar to previous studies (*Aboodarda, Spence & Button, 2015*; *Shu et al., 2021*).

## Adipose tissue thickness

A high-resolution ultrasound (Logiq E, GE healthcare China, Wuhan, China) was used to evaluate subcutaneous adipose tissue thickness (ATT) of the gastrocnemius. A multi-frequency (8–13 MHz) linear probe was used with the frequency set to 12 MHz, gains set to 66 db, and all time gains set to neutral position. Scans were performed longitudinally on the NIRS placement location with the participant lay prone. The obtained images were saved and imported to ImageJ software (*Schneider, Rasband & Eliceiri, 2012*), and ATT was quantified at the center of the image.

## Near-infrared spectroscopy

A continuous wavelength near-infrared spectroscopy (NIRS) device (NIRS; 58* 28* six mm; PortaLite, Artinis Medical Systems B.V., Einsteinweg, The Netherlands) was used for all skeletal muscle metabolic and microvascular reactivity assessment. The NIRS device can be used to quantify the relative concentration changes in oxygenated- and deoxygenated hemoglobin within the measured tissue by resolving the optical absorption characteristics using the modified Beer-Lambert law. This particular NIRS have three separation distances which provide approximal penetration depth of 1 cm, 1.5 cm and 2 cm, respectively. Using the spatially resolved spectroscopy technique, it also provides tissue saturation index (TSI) as a measure for tissue oxygenation status.

With the participant lie on a yoga mat on the ground, the NIRS probe was placed on the gastrocnemius muscle at 1/3 distance between the tibial plateau and the medial malleolus at the most prominent part of the muscle and secured using double-sided and self-adhesive tapes. The placement of the NIRS probe was marked using a permanent marker pen to ensure consistency. A solid rubber pad five cm in height was placed under the participant's foot so the gastrocnemius muscle does not touch the yoga mat. A tourniquet connected to a custom-built rapid inflation device was placed just above the knee. A tourniquet (10.5 cm wide, 80.5 cm width) connected to a custom-built rapid inflation device was placed just above the knee (*Chen et al., 2023*; *Chen & Zhang, 2023*).

The baseline test starts with the participant rest for 5 min. The NIRS device was then turned on to record a 1-minute baseline. Thereafter, the tourniquet was rapidly inflated to

~250 mmHg to provide full vascular occlusion to the leg for 5 min. The tourniquet was then rapidly deflated. The participant rested his leg during the entire test, and NIRS signal was continuously monitored throughout the test until no significant changes to the signal could be observed.

The NIRS device was subsequently removed to allow for proper foam rolling intervention. For skeletal muscle microvascular test following the foam rolling intervention, the NIRS device was re-attached to the same location within 30 s following the last bout of foam rolling. A 15–30 s baseline was collected before the occlusion starts. The time it takes from the foam rolling intervention stops to the start of the occlusion was kept to less than 1 min. The same procedure of tourniquet as baseline was repeated after the foam rolling interventions.

### Data analysis

The NIRS data were analyzed using a custom-written MATLAB script (version 2020b; The Mathworks, Natick, MA, USA). TSI was further processed, as this is an indicator that represents the dynamic balance between the oxygenated and deoxygenated hemoglobin within the skeletal muscle, and less likely to be influenced by ATT (*Barstow, 2019*). TSI slope of decrease between 120 s to 150 s (TSI occlusion slope) during the five-minute occlusion was calculated to determine changes in skeletal muscle oxidative metabolic rate. Three commonly used microvascular reactivity indexes, including the first 10 s slope of increase following tourniquet release (TSI10), the time it takes for TSI to reach half of its peak recovery magnitude (TSI1/2), and TSI peak reactive hyperemia were assessed (*Chen et al., 2023*; *Dellinger, Figueroa & Gonzales, 2023*; *Ihsan, Labidi & Racinais, 2023*).

### Statistics

A priori power analysis was performed to determine sample size using the G*power software (G*Power 3.1; Heinrich-Heine-Universität Düsseldorf, Dusseldorf, Germany) (*Faul et al., 2009*). Given an estimated moderate effect size of 0.35, an alpha level of 0.05 and power (1-$\beta$) of 0.8, it was estimated that 13 participants are required. All data are processed using SPSS 27.0 (IBM Corp., Armonk, NY, USA). Shapiro–Wilk test was used to determine if data follow normal distribution. Baseline measurements of the muscle oxidative metabolic rate and microvascular reactivity were compared using paired $t$-test. All data are processed using SPSS 27.0 (IBM Corp., Armonk, NY, USA). Two-way repeated ANOVA test was used with one repeated factor being the foam rolling method (*i.e.,* with or without vibration), and the other repeated factor being time (*i.e.,* pre- and post-foam rolling intervention). Data are presented as Mean ± SD in tables. Alpha level was set at 0.05.

## RESULTS

The physical characteristics of all participants, including physical activity level and subcutaneous ATT, are summarized in Table 1. All data regarding NIRS measurements were normally distributed. No significant differences were detected for muscle oxidative metabolism ($p = 0.206$) or microvascular reactivity ($p = 0.724$ for TSI1/2, $p = 0.511$ for peak reactive hyperemia and $p = 0.871$ for TSI10) at baseline for the two testing days.

**Table 1  Physical characteristics for all participants.**

|  | Participants ($n = 15$) |
|---|---|
| Height (cm) | 176.2 ± 4.1 |
| Weight (kg) | 70.9 ± 11.0 |
| Age (years) | 21.5 ± 1.2 |
| BMI (kg/m$^2$) | 22.8 ± 2.9 |
| ATT (cm) | 0.6 ± 0.1 |
| Physical activity level (MET-min/w) | 4,261.5 ± 1,890.2 |

Notes.
ATT, adipose tissue thickness.

**Table 2  Statistical results for all NIRS measurements.** TSI occlusion slope (marker for oxidative metabolic rate), tissue saturation index slope of decrease between 120 s to 150 s during the five-minute occlusion. TSI peak, peak reactive hyperemia magnitude following tourniquet release; TSI1/2, the time it takes for TSI to reach half of its peak recovery magnitude; TSI10, the first 10s slope of increase following tourniquet release.

|  | With vibration | | Without vibration | | p | | |
|---|---|---|---|---|---|---|---|
|  | Pre | Post | Pre | Post | Time | Method | Interaction |
| TSI occlusion slope (%/s) | 0.05 ± 0.02 | 0.04 ± 0.01 | 0.06 ± 0.02 | 0.05 ± 0.02 | 0.005** | 0.205 | 0.530 |
| TSI peak (%) | 19.22 ± 4.63 | 20.24 ± 5.08 | 19.77 ± 5.75 | 21.79 ± 8.03 | 0.046* | 0.286 | 0.287 |
| TSI1/2 (s) | 9.33 ± 1.93 | 10.09 ± 1.98 | 9.49 ± 1.71 | 10.01 ± 2.06 | 0.034* | 0.915 | 0.536 |
| TSI10 (%/s) | 1.081 ± 0.34 | 1.049 ± 0.30 | 1.09 ± 0.41 | 1.14 ± 0.49 | 0.963 | 0.469 | 0.200 |

Notes.
*$p < 0.05$.
**$p < 0.01$.

For oxidative metabolic rate related parameters, TSI occlusion slope showed no interaction between foam rolling method and time ($p = 0.530$). However, there was a main effect for time TSI occlusion slope ($p = 0.005$), indicating that both foam rolling and vibration foam rolling induced acute decrease in skeletal muscle metabolism. No main effect for foam rolling method was detected ($p = 0.200$; Table 2).

For microvascular reactivity related parameters, there was no interaction effect ($p = 0.200$), nor was there a foam rolling method ($p = 0.469$) or time ($p = 0.963$) main effect for TSI10. However, while no significant interaction was detected for TSI peak reactive hyperemia ($p = 0.287$), the main effect for time was significant ($p = 0.046$), suggesting a higher peak reactive hyperemia following either foam rolling method. In addition, while no significant interaction was detected for TSI1/2 ($p = 0.536$), there was a main effect for time ($p = 0.034$), suggesting a longer recovery time following either foam rolling method. No significant main effect was detected for foam rolling method for either TSI peak reactive hyperemia or TSI1/2 ($p = 0.286$ and $0.915$, respectively; Table 2).

## DISCUSSION

To the best of our knowledge, this is the first study that evaluated skeletal muscle oxidative metabolism and microvascular reactivity changes in response to vibration foam rolling *vs.* traditional foam rolling. The results indicated that both foam rolling methods acutely reduced skeletal muscle metabolic rate, with significant main effects observed for time
in some microvascular reactivity parameters but no significant interaction between foam rolling method and time for oxidative metabolism-related parameters. For microvascular related parameters, it was shown that both foam rolling methods induced greater peak reactive hyperemia compared to baseline, which is accompanied with longer recovery time as indicated by TSI1/2. However, no change was found for initial recovery speed, as indicated by TSI10. On the other hand, the lack of significant interaction indicates that foam rolling and vibration rolling induced similar microvascular reactivity changes.

In our study, oxidative metabolism was assessed through TSI occlusion slope. Our results showed that both foam rolling methods effectively reduced metabolic rate with no detectable variabilities between methods. The precise cause for the reduced metabolic rate is unclear. It was previously reported that the application of foam rolling has been shown to reduce muscle stiffness and improve tissue perfusion (*Hotfiel et al., 2017*; *Schroeder, Wilke & Hollander, 2021*), which facilitates muscle blood flow and nutrient delivery, thus reduce muscle's oxygen demand.

Finding also indicates that foam rolling altered selected parameters for skeletal muscle microvascular reactivity. Specifically, the total increase in TSI peak reactive hyperemia is accompanied with longer TSI1/2, however TSI10 was not altered. Taken together, these data suggest that foam rolling on the gastrocnemius lead to an increased, prolonged microvascular response, while the initial microvascular response is not substantially altered. Such finding is intriguing and the mechanisms need to be further explored. It is possible that the mechanical pressure applied during foam rolling may temporarily compresses the microvasculature, followed by a compensatory blood flow influx and redistribution within the muscle, thus leading to the prolonged and sustained, but not the immediate changes. Our finding is in line with a previous study which suggested that blood flow is increased following foam rolling (*Hotfiel et al., 2017*). The increase in peak hyperemia is likely attributed to the enhanced endothelial function and vasodilation, which is mediated by the shear stress that foam rolling exerted on the microvasculature, thus induced nitric oxide release (*Okamoto, Masuhara & Ikuta, 2014*). The improvement in microvascular function could indicate more efficient removal of metabolic byproducts and delivery of oxygen as well as other nutrients during the exercise, thus promotes exercise performance and facilitates post-exercise recovery.

The lack of difference in oxidative metabolism and microvascular reactivity between foam rolling with and without vibration is an interesting finding, and it is in contradiction to some previous studies which proposed that vibration foam rolling provides additional benefits compared to traditional foam rolling (*Cheatham, Stull & Kolber, 2019*; *Kasahara et al., 2022a*; *Nakamura et al., 2022*; *Reiner et al., 2021*; *Romero-Moraleda et al., 2019*). However, studies that found vibration foam rolling to be advantageous generally didn't investigate from a metabolism or microvascular perspective. Some scholars have also found that vibration foam rolling convey similar physiological effects compared to tradition foam rolling, which is in line with our findings (*de Benito et al., 2019*; *Kasahara et al., 2022b*; *Nakamura et al., 2023*). Taken together, it is possible that effects of vibration foam rolling depend on the specific performance and physiological variables that's being investigated. The physiological mechanisms underlying the beneficial effects of foam rolling for the

parameters we measured may have already been maximized, thus leaving little room for additive benefits. Future studies are needed to determine such inference is valid.

Our findings are somewhat contradictory to a previous finding, which indicates that rolling massage, as performed by a second individual on the participant, acutely improve skeletal muscle microvascular reactivity as measured in TSI10, but offered no benefits for oxidative metabolism (*Soares et al., 2020*). The discrepancies may be attributed to several factors. First, the previous study evaluated the forearm muscle and the present study evaluated the leg muscle. The varying muscle groups could mean to different metabolic and baseline vascular characteristics, thus lead to the differences in metabolism and microvascular reactivity. In addition, intervention was carried out differently. Our study adopted a self-administration approach, which is more likely to occur in real-world settings. However, the different pressure levels and techniques may also partly contribute to the observed discrepancies.

These findings suggest practical implications, indicating that incorporating foam rolling into pre-exercise routines may optimize muscle function by modulating metabolism and enhancing microvascular reactivity. These effects of foam rolling may efficiently lower metabolic cost thus conserve energy, and help supply nutrients and removing metabolic byproducts which should allow for better performance and post-exercise recovery. The lack of significant difference between foam rolling with and without vibration, suggest for the purposes mentioned above, traditional foam rolling may offer sufficient benefits, and adding vibration does not offer additional advantages.

Limitations of this study must be addressed. First, the vibration intensity was set to medium setting, which corresponds to 60 Hz. It is unclear whether changing the intensity of the roller may induce different metabolic and microvascular responses, which is an area that worth further exploration. Second, the participants were exclusive young healthy, physically active males. Therefore, whether the observed effects could be generalized into other populations with different age, gender or physical activity routine, is currently unknown and needs to the further explored. Third, although the participants in this study were all experienced with foam rolling, and the intensity of rolling was controlled *via* VAS, individual variations in technique may also influence the measurement outcome.

## CONCLUSIONS

In summary, this study compared the effects of traditional foam rolling and vibration foam rolling on skeletal muscle oxidative metabolism and microvascular reactivity in the gastrocnemius muscle. Both methods significantly reduced the metabolic rate and enhanced microvascular reactivity, as indicated by increased peak reactive hyperemia and longer recovery times. However, no significant differences were found between the two foam rolling methods, suggesting that vibration does not provide additional benefits over traditional foam rolling in these aspects.

## ACKNOWLEDGEMENTS

We thank all participants for taking part in this study.

### Funding

This study was funded by the Central China Normal University (Grant NO. 31101222041 and 30106240179) and the National Natural Science Foundation of China (NO. 82301791). The funders had no role in study design, data collection and analysis, decision to publish, or preparation of the manuscript.

### Grant Disclosures

The following grant information was disclosed by the authors:
Central China Normal University: 31101222041, 30106240179.
National Natural Science Foundation of China: 82301791.

### Competing Interests

The authors declare there are no competing interests.

### Author Contributions

- Haizhen Huang conceived and designed the experiments, performed the experiments, analyzed the data, prepared figures and/or tables, authored or reviewed drafts of the article, and approved the final draft.
- Bin Leng performed the experiments, analyzed the data, authored or reviewed drafts of the article, and approved the final draft.
- Chuan Zhang conceived and designed the experiments, prepared figures and/or tables, authored or reviewed drafts of the article, and approved the final draft.

### Human Ethics

The following information was supplied relating to ethical approvals (i.e., approving body and any reference numbers):

Central China Normal University Institutional Review Board (NO. CCNU-IRB-202306005a).

### Data Availability

The data is available at figshare: Zhang, Chuan; Huang, Haizhen (2024). Foam rolling_data.xlsx. figshare. Dataset. Available at https://doi.org/10.6084/m9.figshare.26172322.v1.

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
