# Peer review of "Impact of foam rolling with and without vibration on muscle oxidative metabolism and microvascular reactivity"

_PeerJ, doi:10.7717/peerj.18180_

## Round 0.1 · original submission · Major Revisions

Reviewers found merit in your manuscript but requested revisions. Additional clarity is required in the methods and more background information is necessary.

Reviewer 1 ·

Basic reporting

The manuscript is well-written which makes it easy to understand the rationale behind it as well as the methods used to answer the research question.

Experimental design

The experimental design is my main concern. I understand the idea that self-administration approach is more realistic, however it is not ideal to answer a research question. Individual variations on technique may change blood flow distribution to the limbs and also influence local muscle oxygenation, thereby affecting the NIRS outcomes

Validity of the findings

No comments

Additional comments

No comments

Reviewer 2 ·

Basic reporting

The primary objective of this study was to investigate oxidative metabolism and microvascular reactivity changes in response to vibration foam rolling vs. traditional foam rolling. The two foam rolling methods significantly reduced the metabolic rate and enhanced microvascular reactivity, however, no significant differences were found between the two interventions for these measures. These findings suggest that vibration did not provide additional benefits. The research question is practical; however, there are concerns and questions about the problem statement (specifically, the effectiveness of vibration foam rolling compared to traditional foam rolling), the research method, the interpretation of the findings and the format of the manuscript. Some of these concerns have been outlined in the following sections.

1- Introduction. The problem statement and rationale behind the research hypothesis need to be clarified. Currently, the effectiveness of vibration foam rolling compared to conventional foam rolling has not been established in the Introduction. For instance, in lines 54-55, no study has been cited that would show the additional effect of vibration foam rolling over the traditional method, while multiple studies suggest no additive effect. Also, in lines 63-66 it is stated that “Given the increasing interest in using vibration foam rolling both for post-exercise recovery as well as a warm-up technique…”. However, no study has been cited to show these effects. Please provide direct evidence and references supporting the effect of vibration foam rolling on the warm-up and recovery before and after exercise (please see line 69-70 as well). Without this evidence, search for physiological mechanisms underpinning the additive effect of vibration seems to redundant and unjustified. In line 67, it is stated “In an effort to bridge this gap in the current knowledge…” Please clarify what gap you are referring to. So far, the supporting statements do not show this gap.

2- Introduction. As mentioned above, evidence and rationale should be provided to justify the measurement of oxidative metabolism and microvascular reactivity to the two types of foam rolling. Please expand on lines 67-69 and cite relevant articles.

Experimental design

3- Methods. Please explain how the sample size calculation was performed.

4- Methods. Please explain why females were not included in the study. This limits the results of the study to males only.

5- Methods. Foam rolling necessitates contractions of other muscle groups to perform the task. This can increase whole-body blood flow affecting local blood flow and vascular function. Please explain how the author controlled this effect.
6- Were the data normally distributed? Please report the normality and sphericity of your data.

7- Results. Please provide baseline comparisons for the muscle oxygenation and microvacular reactivity in the two sessions?

Validity of the findings

8- It is stated that “both foam rolling methods effectively reduced metabolic rate… The decreased oxidative metabolism may help conserve energy…” (line 187-189). However, prior studies suggest that foam rolling and rolling massage increase metabolic rate and microvascular reactivity! These findings seem to contradict prior investigations! The authors have cited Madoni et al. (2018) to support the idea that muscle, thus “foam rolling could significantly reduce muscle activation” however the Madoni et a. reported muscle activity during eccentric and concentric contractions. This observation is not applicable to resting condition where the current study was performed at. In addition, Madoni et al., showed the same decline in EMG without Rolling intervention (at Control), that questions the validity of their findings during contraction! Other lines of evidence suggest that an acute use of foam rollers performed immediately prior to running increased cost of running indicating the rolling can negatively affect endurance running performance. These observations refute the your explanation about foam rolling can “reduce energy requirement” (line 194) or “conserve energy” (Line 189).

Additional comments

Specific comments
Lines 40-41. There is no robust evidence that FR or rolling massage can change overall performance. Multiple lines of evidence have rejected this hypothesis. This sentence should be revised.
Line 44. Other researchers before Cheatham & Stull (2020) explored the effect of rolling techniques on pain perception. Please update your citation.
Line 45-46. Pleas provide a reference for the effect of FR or Rolling massage on circulatory system.
Lines 49-50. Is this muscle vibration or tendon vibration? Please clarify. What “similar mechanisms” are involved? Please provide evidence and citations discussing these mechanisms.
Line 50. Please provide references showing this synergistic effect.
Lines 52-54. Please provide direct evidence and references supporting the statement that vibration foam rolling can provide “additional athletic and health-related advantages.”
Line 60. What is the difference between upper vs. lower limbs that could change the outcomes? Please explain the rationale behind the expected difference.
Lines 63-66. As mentioned in the general comments, the “health” and “performance” benefits of foam rolling alone has been very controversial! Please provide evidence and references showing these benefits. Without this evidence, assessing the additional effect of vibration foam rolling on health and performance related benefits seem to be unjustified!
Line 88. The C in the middle of the sentence should be removed.
Line 98-99. The participant attended the lab on two different days and They “had a chance to familiarize with both types of intervention when first reported to the lab, prior to the ultrasound assessment”. They should have performed foam rolling for familiarization. This could have affected the subsequent intervention performed on the same day! That said, a separate familiarization session was required. Please clarify why this was not considered in the methods.
Line 187. Please remove extra t in the middle of the sentence.
Line 130. What type of cuff inflator was used for tourniquet? Please report the size of the cuff. Please clarify what artery was occluded.
Line 140. Please clarify that the same procedure of tourniquet as baseline was repeated after the foam rolling interventions.

·

Basic reporting

The purpose of this paper was to compare the effects of acute foam rolling, with and without vibration, on skeletal muscle metabolism and microvascular reactivity. 15 young males’ gastrocnemius muscle microvascular function were assessed by NIRS with post-occlusive reactive hyperemia technique, before and after foam rolling (with or without vibration). No main effect for condition (FR with vs. without vibration), but there was main effect for time for muscle metabolism and microvascular function parameters. Thus, it was concluded that vibration foam rolling does not offer additional benefits compared to traditional foam rolling alone.

Overall, the flow and language of this paper are easy to follow.

The background of foam rolling with or without vibration can be improved (Lines 58-62). Please add a few sentences here to briefly describe the vibrational FR on physiologic responses, and consider citing the following references:
Chen C-H, Hsu C-H, Chu L-P, Chiu C-H, Yang W-C, Yu K-W, Ye X. Acute Effects of Static Stretching Combined with Vibration and Nonvibration Foam Rolling on the Cardiovascular Responses and Functional Fitness of Older Women with Prehypertension. Biology. 2022; 11(7):1025. https://doi.org/10.3390/biology11071025
Kasahara K, Konrad A, Yoshida R, Murakami Y, Koizumi R, Sato S, Ye X, Thomas E, Nakamura M. Comparison of the Prolonged Effects of Foam Rolling and Vibration Foam Rolling Interventions on Passive Properties of Knee Extensors. J Sports Sci Med. 2022 Dec 1;21(4):580-585. doi: 10.52082/jssm.2022.580. PMID: 36523900; PMCID: PMC9741721.
Nakamura M, Kasahara K, Yoshida R, Murakami Y, Koizumi R, Sato S, Takeuchi K, Nishishita S, Ye X, Konrad A. Comparison of The Effect of High- and Low-Frequency Vibration Foam Rolling on The Quadriceps Muscle. J Sports Sci Med. 2022 Sep 1;21(3):376-382. doi: 10.52082/jssm.2022.376. PMID: 36157391; PMCID: PMC9459764.

Experimental design

Lines 85-92, it is not known if the conditions (with vs without vibration) were randomized. Line 88, check spelling “Cwere”.
The participants information is missing. Please add this section after the “Study Design”. Please also include the sample estimation process so readers will know if the current sample size is enough.
Please also justify why females were not included in this study.
Line 97, why 60Hz? Previous research suggests that lower frequency may offer better results.

Validity of the findings

Although the main findings of the study were not significant (no interaction and no main effect for condition), the result was still interesting to discuss.

---

## Round 0.2 · accepted · Accept

The manuscript is ready for publication.

Reviewer 1 ·

Basic reporting

N/A

Experimental design

N/A

Validity of the findings

N/A

Additional comments

N/A

·

Basic reporting

Thanks for the responses to the comments as well as the revised manuscript. I believe the paper is reaching the publication standard. I have no further questions.

Experimental design

Satisfied

Validity of the findings

The findings appear to be valid.

Additional comments

None